# Performance of Polyvinyl Alcohol/Bagasse Fibre Foamed Composites as Cushion Packaging Materials

**Baodong Liu [1], Xinjie Huang [1], Shuo Wang [1], Dongmei Wang [2],\* and Hongge Guo [1],\***

[1] Department of Packaging Engineering, School of Light Industry Science and Engineering, Qilu University of Technology, Jinan 250353, China; lbd980918@163.com (B.L.); 17854117642@163.com (X.H.); 17854115944@163.com (S.W.)

[2] Packaging Planning and Design, School of Communication, Shenzhen Polytechnic, Shenzhen 518000, China

\* Correspondence: sxxawdm@szpt.edu.cn (D.W.); ghg@qlu.edu.cn (H.G.)

**Abstract:** This work was designed to determine the mechanical properties and static cushioning performance of polyvinyl alcohol (PVA)/bagasse fibre foam composites with a multiple-factor experiment. Scanning electron microscopy (SEM) analysis and static cushioning tests were performed on the foamed composites and the results were compared with those of commonly used expanded polystyrene (EPS). The results were as follows: the materials had a mainly open cell structure, and bagasse fibre had good compatibility with PVA foam. With increasing PVA content, the mechanical properties of the system improved. The mechanical properties and static cushioning properties of the foam composite almost approached those of EPS. In addition, a small amount of sodium tetraborate obviously regulated the foaming ratio of foamed composites. With increasing sodium tetraborate content, the mechanical properties of foamed composites were enhanced. The yield strength and Young's modulus of the material prepared by reducing the water content to 80.19 wt% were too high and not suitable for cushioned packaging of light and fragile products.

**Keywords:** polyvinyl alcohol; bagasse fibre; foamed composites; cushion packaging material



## 1. Introduction

Packaging is an indispensable part of commodity circulation. In the logistics and transportation processes, not only is external packaging needed, but internal packaging is also needed to pack objects or fill gaps [1]. The foaming materials often used in the cushion packaging area can be divided into four categories: organic foam material, plant fiber foam material [2,3], metal foam material [4–7] and ceramic foam material [8,9]. Organic foam material is a closed cell foam made of organic materials with emulsifiers, foaming agents, curing agents and other auxiliaries such as expanded polystyrene (EPS), polyethylene foam (EPE), etc. These are widely used due to their low density, excellent buffer performance and low cost. However, they are hard to biodegrade, thus causing "white pollution". EPS also precipitates toxic substances at high temperatures and harms human health [10,11]. Plant fiber foam is a polymer thermoplastic polymer made from plant fibers (such as orange sticks, seaweed, etc.) and starch (such as rice, corn, potato, etc.), using principles of biological recombination and molecular recombination technology, and special production technology. Research on environmentally friendly cushion packaging materials has therefore become an inevitable trend in the 21st century [12]. In recent years, metal foam material and ceramic foam material are seldom used in packaging.

The main biodegradable packaging materials currently on the market are corrugated cardboard, honeycomb cardboard and paper pulp moulding. Corrugated board and honeycomb paperboard are expensive and exhibit unsatisfactory buffer performance [13,14]. The production process for paper pulp moulding is complex and is only used to make small cushioned packaging material [15]. In summary, all of these buffer packaging materials have some limitations.

Due to their environmental friendliness, low cost, and unlimited market potential, the development of plant fibre cushioning materials has attracted increasing interest. Numerous studies have made great breakthroughs in the development of plant fibre cushioning materials [16–18], and research on plant fibre foaming materials has become a new area of interest. These materials use plant fibre as the main raw material and contain various additives, including foaming agents, nucleating agents, adhesives, foam stabilizers, thickeners, and plasticizers; when mixed in a slurry, the foam is then prepared by moulding foaming, baking foaming, extrusion foaming or microwave foaming, after which the material is finally dried and shaped [15]. The buffer packaging material prepared by this method has the advantages of a simple preparation process, a widely available source of raw materials, and complete susceptibility to degradation, and its mechanical performance is essentially equal to that of EPS. The plant fibre foam material expands spontaneously in response to temperature and pressure. Although it is difficult to obtain materials featuring low density and substantial foaming [19], the density of these fibres is less than 0.1 g/cm$^3$, and the mechanical strength of the material is roughly equal to that of EPS. At present, the density of the most commonly used EPS is between 0.005 and 0.015 g/cm$^3$, which means that there is a large gap between the densities of the two materials [20,21]. To overcome this difficulty, researchers have used various methods to reduce the density of the material, and low-density plant fibre cushion packaging material has become popular in the packaging industry [22,23]. Bagasse fibre is the residue remaining from sugarcane after squeezing [24]. As a natural plant fibre material it can be naturally degraded [25], exhibits a low cost and its output is seven times the total output of jute [26–28], kenaf, and hemp [29]. Polyvinyl alcohol is a polyhydroxy polymer produced by the hydrolysis of polyvinyl acetate, and it exhibits good mechanical properties [30,31]. It is nontoxic, biocompatible, water soluble, semicrystalline, and fully biodegradable, and it has a relatively lower cost than the currently dominant foam precursor, polyurethane [32]. Bagasse fibre and polyvinyl alcohol meet the requirements of environmental protection.

In this work, the components of foamed composites were PVA and bagasse fibre, and the production process utilized a nucleating agent and a cross-linking agent, with water as a plasticizer. The foamed composites used as cushion packaging were prepared by the mechanical foaming method. The advantage of composite materials is that the components can complement each other [33] and produce synergies that avoid the high-density defects of plant fibre foaming material while providing good mechanical performance. Through multifactorial experiments the mechanical properties, static cushioning properties, and variation laws of the microstructures of these materials were studied via single factor changes (content of PVA, sodium tetraborate and water).

## 2. Experiments

### 2.1. Experimental Materials

Polyvinyl alcohol was purchased from Shanghai Aladdin Biological Technology Co., Ltd. (Shanghai, China), specifically chemically pure PVA 1788; the degree of polymerization was 1700 and the degree of alcoholysis 88%, while the average Mw was 85,000–124,000; it had good water solubility and dissolved quickly in cold or hot water. Bagasse fibre (chemical pulp) was supplied by Nanning Sugar Industry Co., Ltd. (Nanning, China). The nucleating agent and sodium tetraborate (analytically pure) were obtained from Tianjin Bodi Chemical Co., Ltd. (Tianjin, China). Deionized water was prepared with filtering equipment in our laboratory.

### 2.2. Experimental Apparatus

Experiments involved the use of an electronic balance (Accuracy: 0.001 g, Mettler Toledo Co., Ltd., Greifensee, Switzerland), a mechanical stirrer (stepless speed change, IKA RW20 Digital, Shanghai Yikong Mechanical and Electrical Co., Ltd., Shanghai, China), a thermostat water bath (HH-2, Changzhou Putian Instrument Manufacturing Co., Ltd., Changzhou, China), a vacuum freeze dryer (74200-30, Labconco Co., Ltd., Kansas, MO,

USA), an electric blast drying oven (DHG-9140A, Shanghai Jinghong macro laboratory equipment Co., Ltd., Shanghai, China), a universal testing machine (M-3050, Shenzhen Reger Instrument Co., Ltd., Shenzhen, China), and a scanning electron microscope (JCM6000, Japanese Electronics Maker Hitachi Ltd., Tokyo, Japan).

### 2.3. Specimen Preparation

First, bagasse cellulose was blended with polyvinyl alcohol solution at room temperature. Then, calcium carbonate and sodium tetraborate were added in turn. After mixing for 5 min at a speed of 1600 r/min, the mixture was cast in the cube mould (length × width × height = 5 cm × 5 cm × 4 cm) and put into the refrigerator at low temperature to freeze for 12 h. Finally, it was dried and shaped in a vacuum freeze dryer. Formulations are listed in Tables 1–3.

**Table 1.** Formulations of composites with different PVA contents.

| Specimens | PVA (g) | Fibre (g) | Water (g) | Sodium Tetraborate (g) | Nucleating Agent (g) |
|---|---|---|---|---|---|
| 1: PVA of 7.11 wt% | 5 | 5 | 60 | 0.06 | 0.3 |
| 2: PVA of 8.41 wt% | 6 | 5 | 60 | 0.06 | 0.3 |
| 3: PVA of 9.67 wt% | 7 | 5 | 60 | 0.06 | 0.3 |
| 4: PVA of 10.91 wt% | 8 | 5 | 60 | 0.06 | 0.3 |
| 5: PVA of 12.10 wt% | 9 | 5 | 60 | 0.06 | 0.3 |

**Table 2.** Formulations of composites with different sodium tetraborate content.

| Specimens | Sodium Tetraborate (g) | PVA (g) | Fibre (g) | Water (g) | Nucleating Agent (g) |
|---|---|---|---|---|---|
| a: sodium tetraborate of 0 wt% | 0 | 8 | 5 | 60 | 0.3 |
| b: sodium tetraborate of 0.027 wt% | 0.02 | 8 | 5 | 60 | 0.3 |
| c: sodium tetraborate of 0.055 wt% | 0.04 | 8 | 5 | 60 | 0.3 |
| d: sodium tetraborate of 0.082 wt% | 0.06 | 8 | 5 | 60 | 0.3 |
| e: sodium tetraborate of 0.109 wt% | 0.08 | 8 | 5 | 60 | 0.3 |

**Table 3.** Formulations of the composites with different water contents.

| Specimens | Water (g) | PVA (g) | Fibre (g) | Sodium Tetraborate (g) | Nucleating Agent (g) |
|---|---|---|---|---|---|
| I: Water of 80.19 wt% | 54 | 8 | 5 | 0.04 | 0.3 |
| II: Water of 80.76 wt% | 56 | 8 | 5 | 0.04 | 0.3 |
| III: Water of 81.30 wt% | 58 | 8 | 5 | 0.04 | 0.3 |
| IV: Water of 81.81 wt% | 60 | 8 | 5 | 0.04 | 0.3 |
| V: Water of 82.29 wt% | 62 | 8 | 5 | 0.04 | 0.3 |
| VI: Water of 82.75 wt% | 64 | 8 | 5 | 0.04 | 0.3 |
| VII: Water of 83.19 wt% | 66 | 8 | 5 | 0.04 | 0.3 |

*2.4. Testing and Characterization*

2.4.1. Static Compression Properties

Static compression experiments were performed with a microcomputer-controlled electronic universal testing machine. After the material was formed, the temperature and humidity of the samples were established according to ISO 2233-2000; the compression performance was tested according to the GB/T 18942.1-2003: static compression test method of cushioning materials for packaging at a compression speed of 12 mm/min. Each sample was subjected to three repeated experiments. The results presented are the average values of the three experiments. The material load-displacement curve was obtained, and the stress-strain curve of the material was calculated according to the sample size. The samples had almost the same length × width (5 cm × 5 cm); the thickness was around 3 cm.

2.4.2. Yield Strength and Young's Modulus

The yield stress value was regarded as the compression strength ($\delta_*$) when the corresponding strain was 10%. The Yield modulus is the compression strength ($\delta_*$)/10%.

The formula for Young's modulus is as follows:

$$E_{sj} = \frac{1}{2}\left(\frac{\delta_{j+1} - \delta_j}{\varepsilon_{j+1} - \varepsilon_j} + \frac{\delta_j - \delta_{j-1}}{\varepsilon_j - \varepsilon_{j-1}}\right) \tag{1}$$

where $\delta$ is the stress of the material and $\varepsilon$ is the strain of the material.

The modulus variance of the material with strains between 0.05–5% is the smallest, and the corresponding modulus is the Young's modulus of the material.

2.4.3. Cushion Coefficient

The energy absorption formula for the material is

$$e = \int_0^\varepsilon \delta d\varepsilon \tag{2}$$

where $e$ is the energy absorbed by the material, that is, the area between the stress-strain curve and the *x*-axis. The cushion coefficient formula is

$$C = \frac{\delta}{e} \tag{3}$$

where *C* is the cushion coefficient.

### 2.4.4. SEM Analysis

The external and internal structures of the material were observed by scanning electron microscopy (JCM6000). The material was first cut into slices approximately 2 mm thick and then processed by spraying gold. Finally, the samples were studied with an accelerating voltage of 15 kV. Samples for which the PVA content was 10.91 wt% were used for surface SEM analysis at magnifications of 30, 90, 150 and 300×. Samples with a PVA content of 8.41%, 9.67% and 10.91% were used for SEM analysis of internal sections at magnifications of 30, 50, 90, 300 and 500×.

## 3. Result and Analysis
### 3.1. Effect of PVA Content on Material Properties
### 3.1.1. SEM Analysis

The surface morphology of the materials when the PVA content was 10.91 wt% is shown in Figure 1, with magnifications of 30, 90, 150 and 300×. It is obvious that the fibres were staggered and evenly distributed in the mixture, forming a support system for the skeleton structure. In Figure 1d, the inside of the cell can be clearly observed, so a preliminary estimate suggests that the material has an open cell structure. Bagasse fibres were distributed uniformly in PVA without agglomeration on the surface.

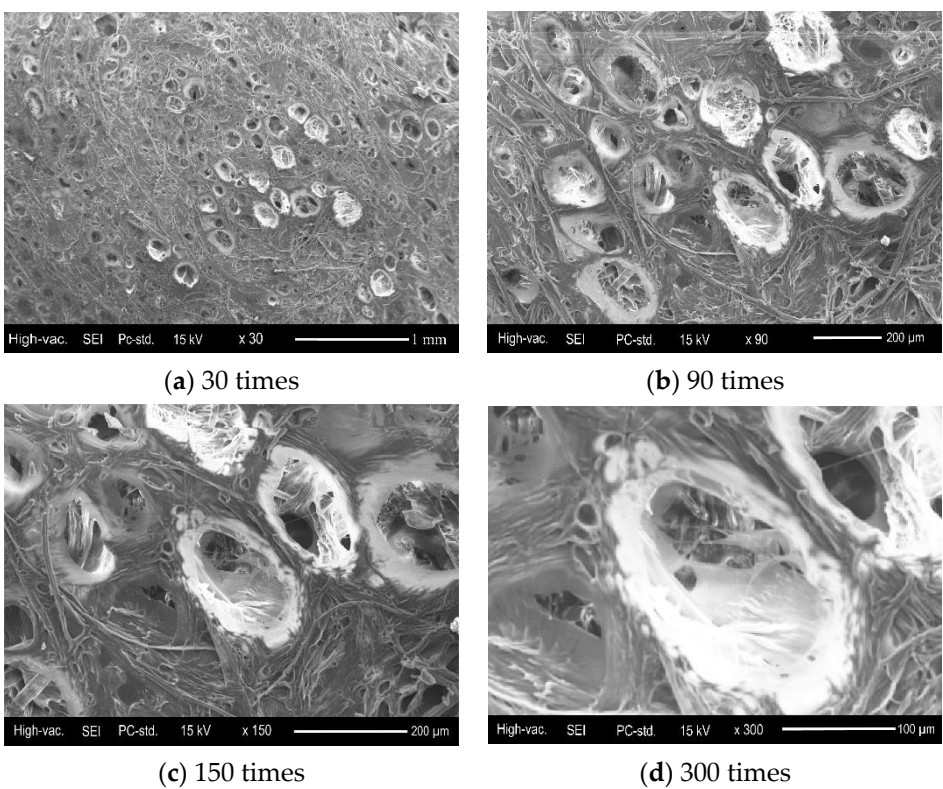

(**a**) 30 times

(**b**) 90 times

(**c**) 150 times

(**d**) 300 times

**Figure 1.** Morphology of the surface of a sample (PVA content of 10.91%) at different magnifications. (**a**) 30 times; (**b**) 90 times; (**c**) 150 times; (**d**) 300 times.

Figure 2 shows that with continuous increases in PVA content, the quantity of cells gradually increased, cell diameters gradually decreased, cell walls gradually thickened, and the macroscopic observation was that the strength of the material increased. The cells in the material consisted of two parts. The main part consisted of the voids formed by air bubbles in the process of mechanical stirring, and the relatively smaller part formed because of the evaporation of water in the drying process.

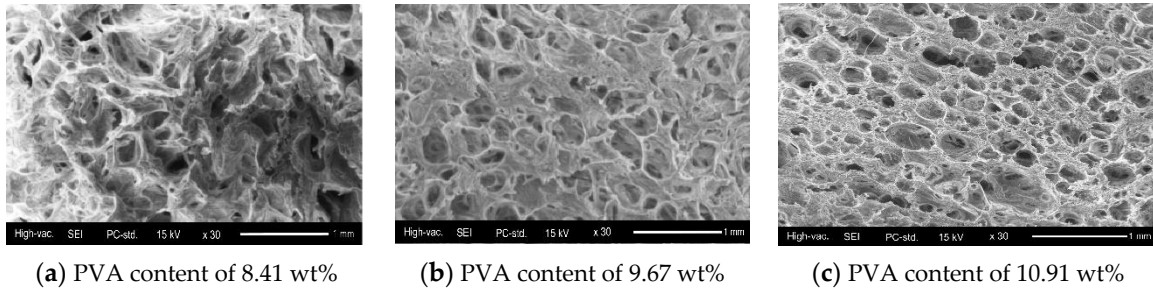

(**a**) PVA content of 8.41 wt%     (**b**) PVA content of 9.67 wt%     (**c**) PVA content of 10.91 wt%

**Figure 2.** Morphologies of internal sections of the samples at 30× magnification. (**a**) PVA content of 8.41 wt%; (**b**) PVA content of 9.67 wt%; (**c**) PVA content of 10.91 wt%.

As shown in Figure 3a, the distribution of cells on the surface of the material was relatively sparse. Figure 3d shows a cell interior through a channel with surface bubble cracks, indicating that the material has an open structure. The vesicles are round or oval in shape. The interface between bagasse fibre and polyvinyl alcohol cannot be clearly distinguished from an interrupted surface in the figure. Only the fibre extends to the outside of the composite from the materials inside the cell, indicating that bagasse fibre and polyvinyl alcohol have good compatibility and are closely combined with each other.

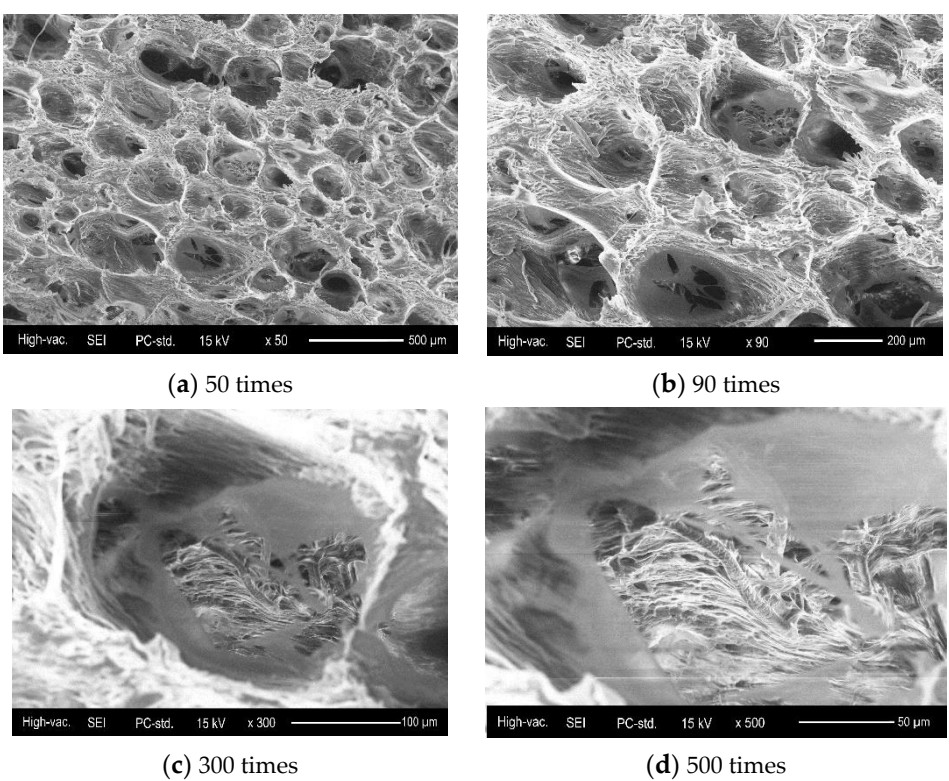

(**a**) 50 times          (**b**) 90 times

(**c**) 300 times         (**d**) 500 times

**Figure 3.** Morphology of an internal section of a sample (PVA content of 10.91%) with different magnifications. (**a**) 50 times; (**b**) 90 times; (**c**) 300 times; (**d**) 500 times.

### 3.1.2. Stress and Strain Curve Analysis

Figure 4 shows the stress and strain curves of different samples. Samples 1–5 contained PVA contents ranging from 7.11 wt% to 12.10 wt%, as listed in Table 1. Sample 6 comprised high-density polystyrene (HDEPS, with a density of 0.015 g/cm$^3$) and sample 7 comprised low-density polystyrene (LDEPS, with a density of 0.005 g/cm$^3$). HDEPS and LDEPS are high foaming materials.

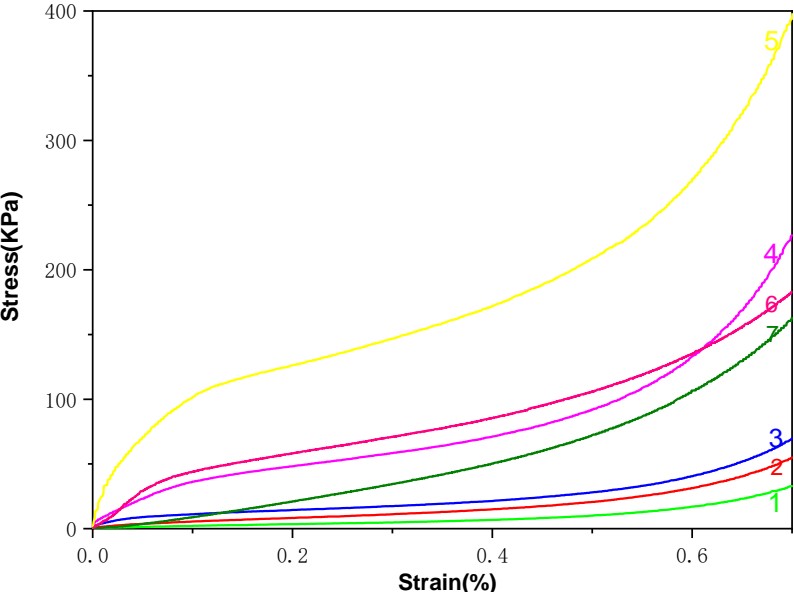

**Figure 4.** Stress and strain curves for materials with different PVA contents.

Sample 5 showed a higher stress and strain curve than HDEPS and LDEPS. The stress for sample 4 was higher than that for LDEPS and similar to that for HDEPS. After entering the densification stage (strain exceeding 0.6), the stress of the composite gradually surpassed that of the HDEPS because the density of the composite is three times larger than that of HDEPS. Therefore, composites first entered the compaction stage, and the stress-strain curve of the material was determined by the characteristics of the material itself rather than by elastic compression of the cell. When the PVA content was less than that in sample 4, the overall strength of the material was gradually lower than that of LDEPS, which indicates that more than 9.67 wt% PVA should be added in this formula to produce a relatively higher stress and strain curve.

### 3.1.3. Analysis of Yield Strength and Young's Modulus

Figure 5 shows that with increasing PVA content, the yield strength and Young's modulus of the material gradually increased, indicating that when PVA was at a low level in these formulas the viscosity of the material had reached the requirements for maintaining cell stability.

The increase in PVA content increased the viscosity of the material, increased the resistance to cell growth, and finally increased the thicknesses of cell walls. In this way, the bearing capacity of the material was stronger. On the other hand (as indicated by the SEM analysis of the material), the higher the PVA content was, the finer the cell of the foamed composite. When stress concentration occurred during the compression process the propagation of the crack tip was effectively hindered with higher PVA content. As greater stress was required to destroy the cell structure, the mechanical strength of the material was better. As can be seen from Figure 6, the crosslinking of PVA with sodium tetraborate increased the molecular weight of PVA and the spatial network structure of PVA was formed, thus the strength of the matrix was greatly improved. This indicates that such crosslink can prevent propagation of the crack tip [34].

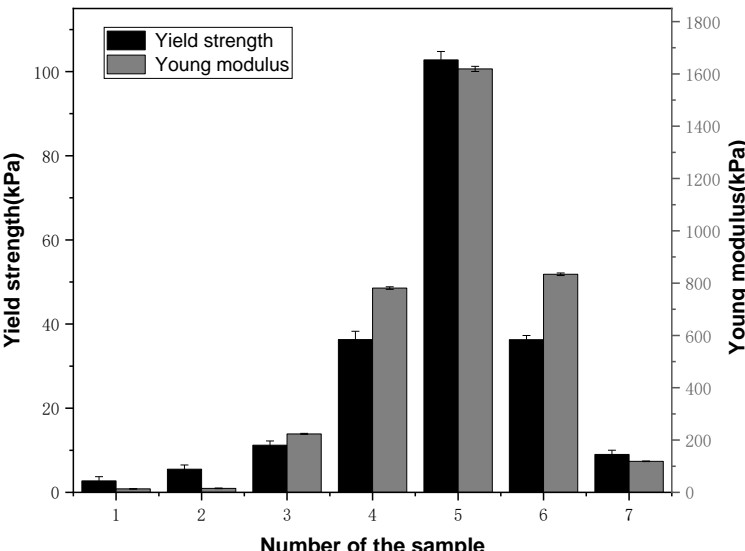

**Figure 5.** Yield strength and Young's modulus of materials with different PVA contents.

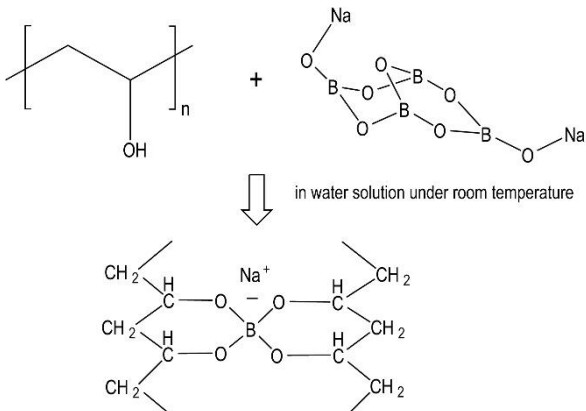

**Figure 6.** Crosslinking reaction equation of PVA.

When the PVA content was 12.10 wt%, the Young's modulus and yield strength of the composite were much higher than those of HDEPS and LDEPS. When PVA content was 9.67%, they remained higher than those of LDEPS, and similar to those of HDEPS. The yield strength and the Young's modulus of the material exhibited the same change trend because the larger the Young's modulus was, the faster the material stress rose in the online elastic stage, and the higher the yield point of the material was. This trend fully explains the wired elastic stage and yield platform stage in the static compression curve of the material.

### 3.1.4. Cushion Coefficient Analysis

Figure 7 shows that when the PVA content was at a low level, the cushion coefficient of the material quickly reached the lowest point with increasing strain, after which the cushion coefficient of the material increased rapidly. At a high stress level, the cushion coefficient was high and the energy absorption level was low, indicating that although the foaming ratio of the material was high, the mechanical strength was poor; thus the material was only suitable for use as a cushioning packaging filler for special lightweight materials.

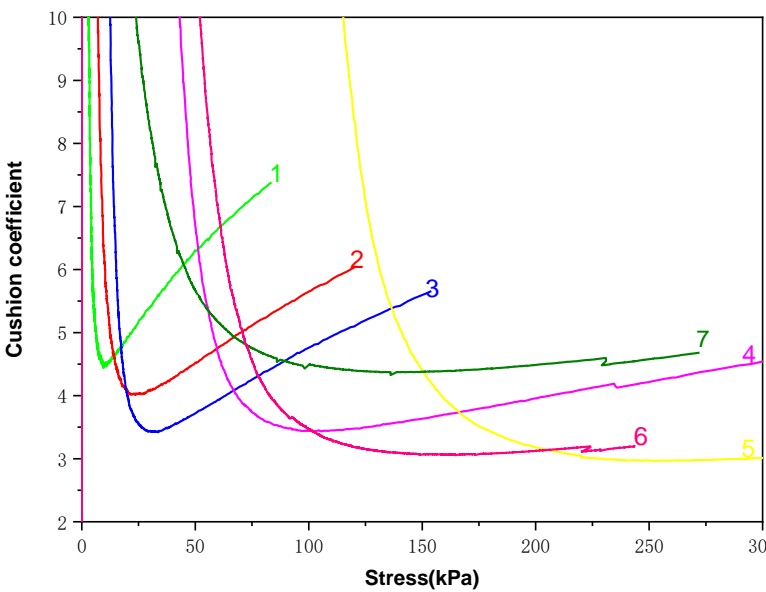

**Figure 7.** Cushion coefficient of materials as a function of stress, with different PVA contents.

When the PVA contents were 10.91 wt% and 12.10 wt%, the minimum cushion coefficients of the material were 3.43 and 2.96, respectively. After the cushion coefficient reached the lowest point, the values did not rise rapidly. The material still had a low cushion coefficient at a higher stress level and exhibited a good energy absorption level.

### 3.2. Effect of Sodium Tetraborate Content on the Mechanical Properties of Materials

Samples a–e contain sodium tetraborate contents ranging from low to high, as listed in Table 2. Sample f is high-density polystyrene (HDEPS, with a density of 0.015 g/cm$^3$) and Sample g is low-density polystyrene (LDEPS, with a density of 0.005 g/cm$^3$). As shown in Figure 8, with continuous increases in sodium tetraborate content the stress-strain level of the material continued to improve, indicating that the added sodium tetraborate served as a crosslinker and increased the degree of PVA crosslinking and improved the viscosity of the system, increasing cell growth resistance, reducing the foaming ratio of the material, and enhancing the relevant mechanical properties. When the sodium tetraborate content exceeded 0.082 wt% (curve d), the stress-strain curve of the material was higher than that of HDEPS.

Figure 9 shows that with increasing sodium tetraborate content the yield strength and Young's modulus of the material gradually increased. During processing, the addition of sodium tetraborate increased the viscosity of the material and regulated the foaming ratio of the foaming material. When the amount of sodium tetraborate reached 0.109 wt% (sample e), stirring of the material became very difficult. If sodium tetraborate was added continuously, the foaming rate suddenly decreased. As the viscosity of the system increased, stirring aggravated the reaction of PVA and sodium tetraborate, further gel reactions occurred, and the walls of the cells disappeared. This resulted in high density and a sudden decrease in the foaming ratio.

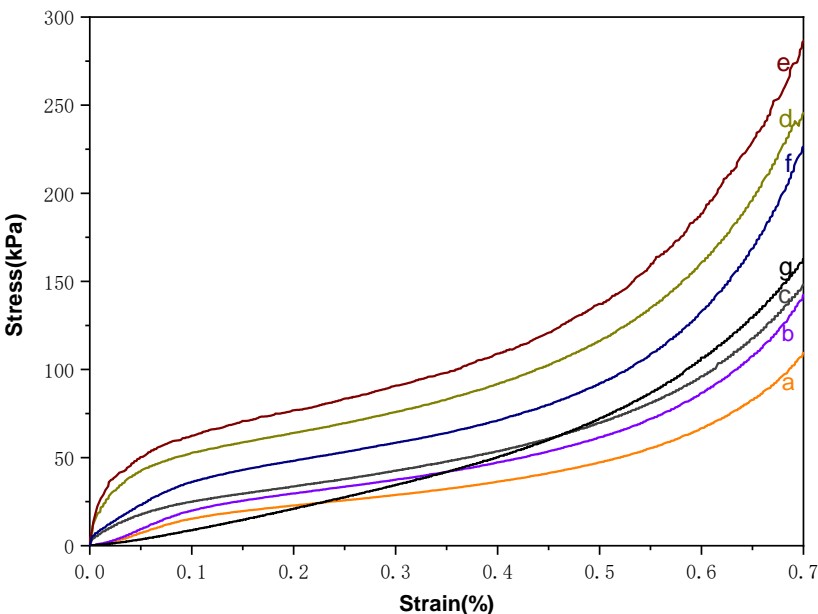

**Figure 8.** Stress and strain curves of materials with different sodium tetraborate contents.

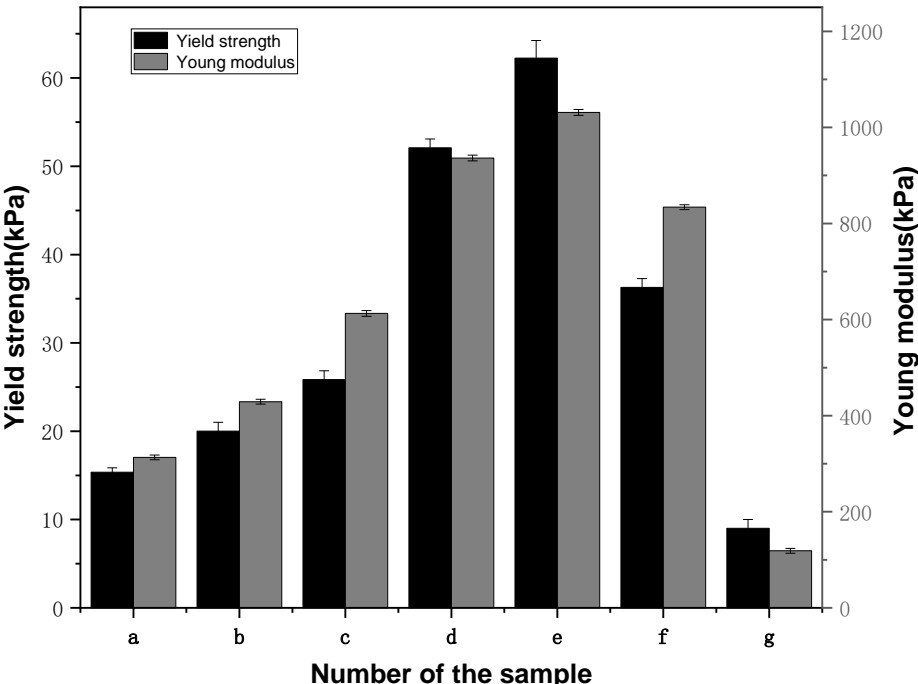

**Figure 9.** Yield strength and Young's modulus for materials with different sodium tetraborate contents.

In Figure 10, the curves for samples a–c exhibited trends relatively similar to that of LDEPS, which indicated that the lower cushion coefficient was obtained when the stress was low, and the minimum cushion coefficient was obtained when the stress was between 50–100 kPa. The material was therefore suitable for cushion packaging of light materials, and the energy absorption capacity was higher than that of LDEPS. When the stress increased, the cushion coefficient rose more rapidly than that of LDEPS, which means that the energy absorption capacity was relatively poor. At this point the composite entered the compaction stage and the buffer capacity was weakened.

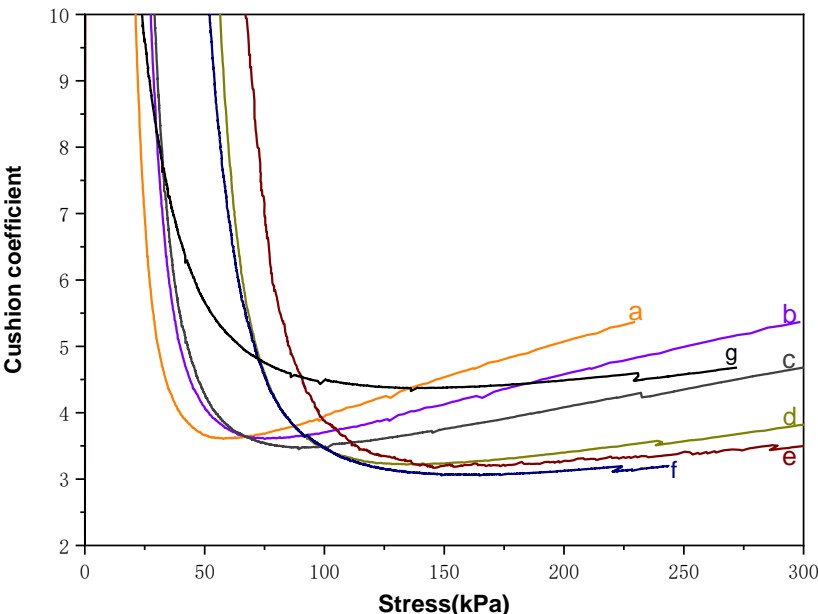

**Figure 10.** Cushion coefficient of materials with different sodium tetraborate contents.

The curves for samples d and e were very close to that of HDEPS, and the cushion coefficients of the materials did not increase rapidly after reaching the minimum point, indicating that the composite was in the yield platform stage like HDEPS, and had good energy absorption capacity.

### 3.3. Effect of Water Content on the Mechanical Properties of Materials

Samples I–IX contained water contents ranging from low to high, as listed in Table 3. Sample VIII was HDEPS and Sample IX was LDEPS. Figure 11 shows that increasing water content meant that the Young's Modulus from the stress-strain curve dropped down and the viscosity of the mixture decreased continuously; moreover, the foams were softer, absorbed compressive energy, and deformed.

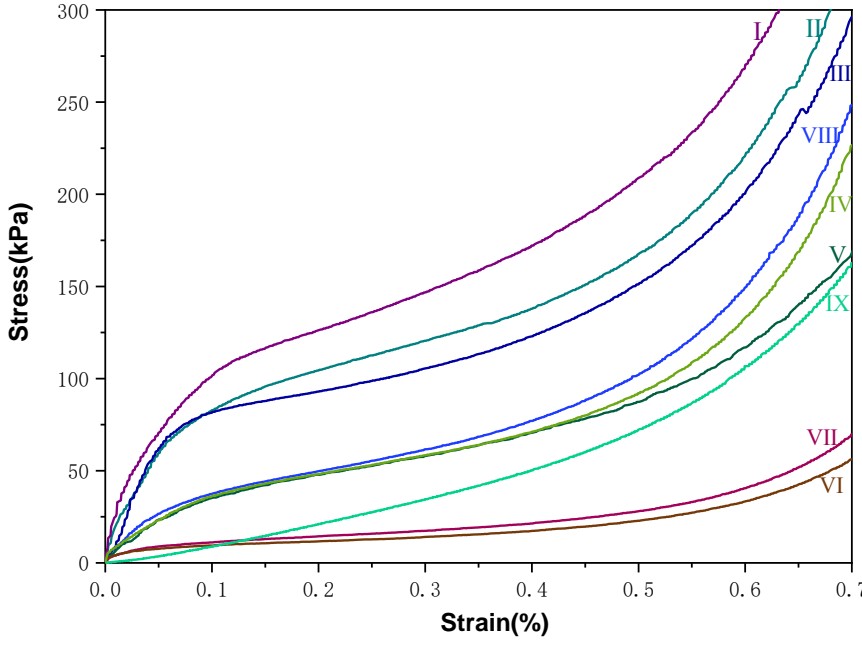

**Figure 11.** Stress and strain curves of materials with different water contents.

Figure 12 shows that with increasing water content, the yield strengths and Young's moduli of the materials decreased. The Young's modulus of sample I was 1920.37 kPa, 130.21% higher than that of sample VIII (HDEPS), indicating that the foamed material made with this water content was relatively hard, and that increasing the water content was conducive to reducing the hardness.

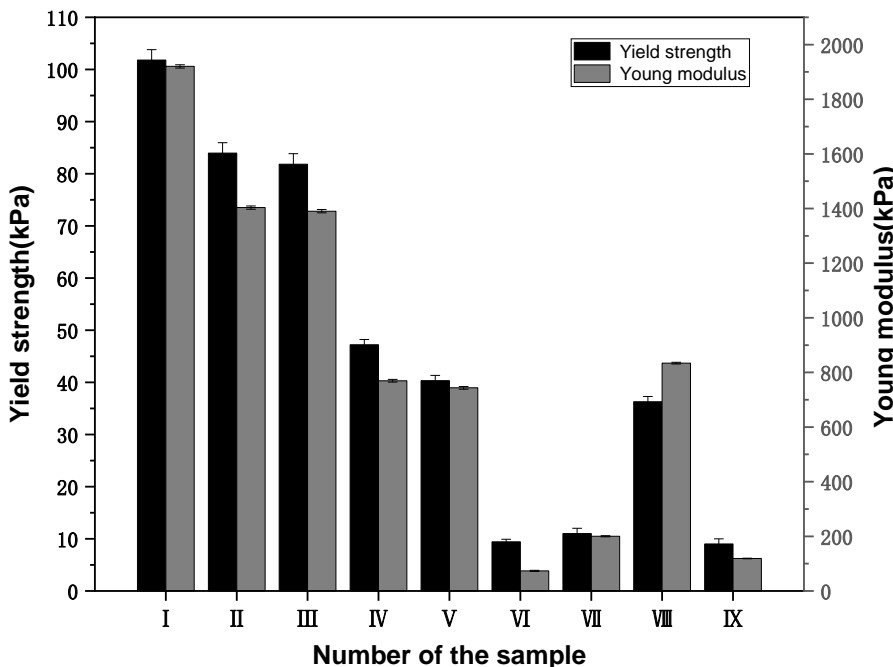

**Figure 12.** Yield strengths and Young's moduli of materials with different water contents.

As shown in Figure 13, with continuous increases in water content, the stress at which materials reached a lower cushion coefficient gradually decreased. Sample I had a low water content, and the stress at the low point for C was too high and not suitable for cushioning protection of lightweight items.

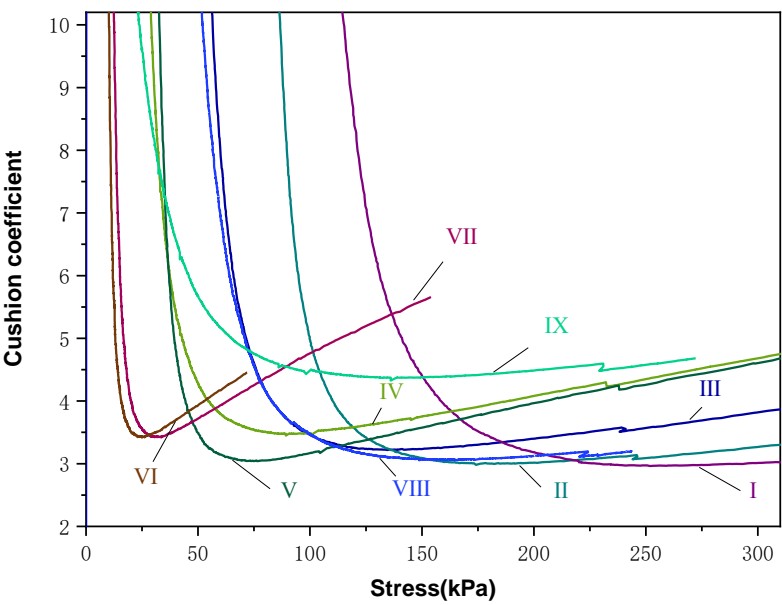

**Figure 13.** Cushion coefficient of materials with different water contents.

## 4. Conclusions

Foams have an open cell structure, and bagasse fibre has good compatibility with PVA. With increasing PVA content, the number of cells in the material gradually increases, the diameters gradually decrease, the cell walls gradually thicken, and the macroscopic observation is that the mechanical properties of the material are enhanced. When the PVA content is 10.91 wt%, the comprehensive mechanical properties and static cushioning properties of the material are closest to those of HDEPS.

When the content of sodium tetraborate constitutes 0.027 wt% or 0.055 wt% of the basic formulation, the mechanical properties of the material are similar to those of LDEPS, and when the content of sodium tetraborate is 0.082 wt% or 0.109 wt%, the mechanical properties of the material are similar to those of HDEPS. The added amount cannot exceed 0.109 wt%, or the degree of gelation is high and not conducive to cell stability.

As a plasticizer, water plays a role in adjusting the hardness and cushioning of foaming materials. If the amount of water added is too small, the material is too hard, and if the amount of water added is too large, the cell strength of the material is low and the deformation is large.

With an appropriate formula, we can obtain PVA/bagasse fibre cushioning materials suitable for replacement of LDEPS or HDEPS. The new foamed material exhibits the characteristics of environmental protection, degradability and low price.

In the future, the dynamic cushioning properties of these foams should be analysed to complete performance simulations under actual conditions. Both matrix materials contain hydroxyl groups. Under high humidity conditions, hydroxyl groups absorb water, resulting in a decline in cushioning performance. In addition, a suitable waterproof coating is needed to protect the cushioning material.

**Author Contributions:** B.L.: Writing-original draft, Investigation, Data curation; X.H.: Software, Visualization; S.W.: Validation, Data curation; D.W.: Conceptualization, Formal analysis; H.G.: Resources, Supervision, Data curation. All authors have read and agreed to the published version of the manuscript.

**Funding:** This research was funded by Qilu University of Technology (Shandong Academy of Sciences) International Cooperation Research Special Fund Project (QLUTGJHZ2018028). Projects funded by Shenzhen Science and Technology Plan International Cooperation Fund (GJHZ20180928161004981).

**Institutional Review Board Statement:** Not applicable.

**Informed Consent Statement:** Not applicable.

**Data Availability Statement:** Not applicable.

**Conflicts of Interest:** The authors declare no conflict of interest.

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
