# Peer review of "Performance of Polyvinyl Alcohol/Bagasse Fibre Foamed Composites as Cushion Packaging Materials"

_coatings, doi:10.3390/coatings11091094_

Round 1
Reviewer 1 Report
This manuscript presents a good planned research. In the implementation there are problems, and the demonstration of the research is not acceptable too.
In the line 33 the authors use this reference: {Aboura, 33 2004 #7}, please change it! Maybe they think of [4] reference.
The specifying of experimental apparatus in 2.2 section is unusual. Modify it!
The specifying of specimen preparation in 2.3 section is incomplete, please give the size of mould and the cutting method of specimens.
The describing of testing and characterization in 2.4 section is incomplete, please give the exact specification of every test method. What type Yield strength and Young modulus were measured: compression or tensile? What was the size of the applied specimen?
The used sample numbering confusing. Why did the author use everywhere sample name as arabic number? Why did not the authors use different sample name, for example: in the first case from sample 1 to sample 5, in the second case from sample a to sample f and in the third case sample I to sample XI? It would be better if the sample name was appear in the Table 1 to 3. Please modify it, and make the labelling clearer.
In the figure 5, 8 and 11 the author used wrong denomination. They used Yield modulus instead of Yield strength. Change it!
In the introduction the authors highlighted the importance of the density of the packaging materials, but during the research the authors did not measure the density of the developed materials. Maybe it would be great the comparison between developed materials and commercial materials according to density-specific properties too.
Reviewer 2 Report
Please add some data about PVA properties. This was missing in the description.
Fig. 5 incorrectly describes the axes. It does not correspond to the description of the drawing.
There is no standard deviation in the charts 5,8,11. Please complete this.
Fig. 7 - in which units we express the deformation, add in Fig.
Reviewer 3 Report
1) Introduction section lacks more in depth analysis of what has been already achieved by other authors in the same or similar fields.
2) Experimental part lacks of information about the size of samples, conditioning and testing conditions for samples.
3) Measurements for closed/open cell content, cell wall thickness and the average cell size are missing. It would not be the case, but the article makes strong statements without any proofs.
4) There is no discussion part. None of the results are compared to other authors works. There are almost no explanations why one or another phenomena happened, no references to assumptions or conclusions.
5) Authors write that PVA content increased the viscosity...I can't see the measurements.
6) The reference list shows the low quality of the paper. Please expand it.
Reviewer 4 Report
The manuscript entitled “Performance of polyvinyl alcohol/bagasse fibre foamed composites as cushion packaging materials” was reviewed. The following comments are necessary to be done:
- Introduction: The authors missed the metal foam in their Introduction. They must provide some literature review on cushion properties of metal foams as one of the most important energy absorber groups: To this end, the research works by Movahedi et al, Orbulov et al and Duarte et al are recommended.
- The markers on SEM images should be enlarged.
- Line 181: “…the stress-strain curve of the material was generally lower than that of LDEPS…”
The above sentence should be corrected to the “…overall strength of the material was gradually lower…”
- The difference between the “Yield Modulus” and “Young Modulus” must be described. How was Yield Modulus calculated?
- Line 195: The authors must provide sufficient discussion on the following sentence:
“When stress concentration occurred during the compression process, the propagation of the crack tip was effectively hindered with higher PVA content”.
- In Figures 4 and 7, the stress-strain curves must be plotted using different line types.
- Line 261: “… Figure 10 shows that increasing water content meant the stress-strain curve of the material was close to the x-axis…”
This writing is not scientific. The authors should focus on the mechanical properties such as Young’s Modulus instead of writing close to X-Axis.
Round 2
Reviewer 1 Report
The quality of the revised version of the manuscript is better, but there are some parts which have to modify:
- in the line 99 tha authors wrote: "... (length* width*height=5cm*5cm*4cm) ..." it would be correct if use "x" instead of "*",
- in the line 117 there is the same fault,
- it would be better, if the markers of samples (1-7; a-f and I-IX) would be given in the Table 1 to 3 which could help the identification of the samples.
Reviewer 4 Report
The comments are addressed.
Author Response
Thank you very much for your affirmation!
This manuscript is a resubmission of an earlier submission. The following is a list of the peer review reports and author responses from that submission.